# LLM-based Skill Diffusion for Zero-shot Policy Adaptation

**Woo Kyung Kim**[1], **Youngseok Lee**[2], **Jooyoung Kim**[1], **Honguk Woo**[1*]

[1] Department of Computer Science and Engineering, Sunkyunkwan University
[2] Department of Electrical and Computer Engineering, Sungkyunkwan University
{kwk2696,yslee.gs,onsaemiro,hwoo}@skku.edu

## Abstract

Recent advances in data-driven imitation learning and offline reinforcement learning have highlighted the use of expert data for skill acquisition and the development of hierarchical policies based on these skills. However, these approaches have not significantly advanced in adapting these skills to unseen contexts, which may involve changing environmental conditions or different user requirements. In this paper, we present a novel LLM-based policy adaptation framework LDuS which leverages an LLM to guide the generation process of a skill diffusion model upon contexts specified in language, facilitating zero-shot skill-based policy adaptation to different contexts. To implement the skill diffusion model, we adapt the loss-guided diffusion with a sequential in-painting technique, where target trajectories are conditioned by masking them with past state-action sequences, thereby enabling the robust and controlled generation of skill trajectories in test-time. To have a loss function for a given context, we employ the LLM-based code generation with iterative refinement, by which the code and controlled trajectory are validated to align with the context in a closed-loop manner. Through experiments, we demonstrate the zero-shot adaptability of LDuS to various context types including different specification levels, multi-modality, and varied temporal conditions for several robotic manipulation tasks, outperforming other language-conditioned imitation and planning methods.

## 1 Introduction

Skill-based learning has demonstrated its potentials in generalizing to novel downstream tasks by leveraging pre-trained skills learned from the offline dataset. Furthermore, the integration of skill-based learning and natural language realizes the remarkable ability to perform practical tasks via the provision of a human-oriented interface where agents are controlled by instructions describing the goals of the task. Building upon this notion, several previous studies have investigated bridging the gap between the human instructions and physical world manipulation by learning semantically meaningful skills given the language-annotated dataset [1, 2, 3]. However, due to the inherently open-ended nature of language, it is impractical to obtain a dataset annotated with a sufficiently wide range of contexts, encompasing various environmental conditions and user requirements, to develop a versatile language-conditioned policy capable of accommodating such diverse contexts. Consequently, as shown in the left side of Figure 1, these prior works are limited to processing the narrow scope of instructions that primarily convey only the goal of the task without any contextual information (presented as case 1).

To address the challenges in language-conditioned skill learning, we explore large language model (LLM)-based policy adaptation approaches that enable zero-shot adaptation to contexts specified in

---

*Honguk Woo is the corresponding author.

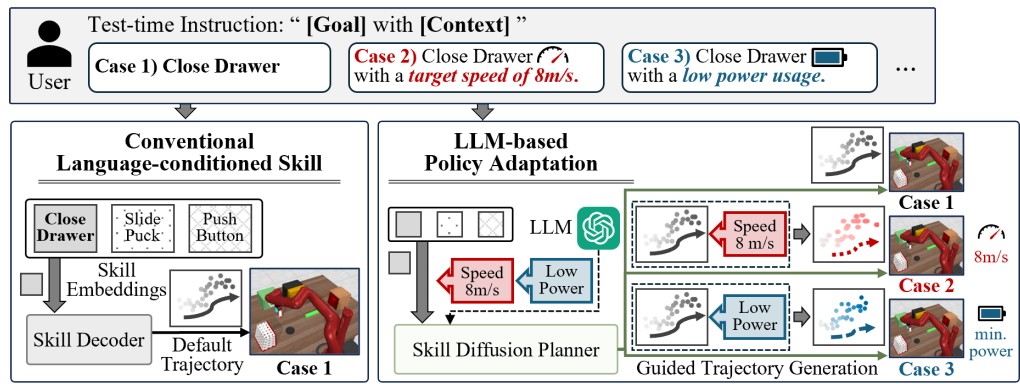

Figure 1: Zero-shot policy adaptation to contexts: In case 1, the instruction includes only the task goal. In cases 2 and 3, the instruction is supplemented by the task goal with the context. Conventional language-conditioned skill approaches struggle to generate trajectories well aligned with the contexts, and typically succeed only for instructions as in case 1. Conversely, our LLM-based policy adaptation approach effectively adapts to the contexts in a zero-shot manner across all cases.

language at test-time. Leveraging the effectiveness of diffusion models in controlling their generation process via loss functions [4], and the code generation capability of LLMs [5], we adapt diffusion with guidance by the loss function generated through LLMs. As illustrated in the right side of Figure 1, our LLM-based policy adaptation is capable of adapting to diverse contexts in a zero-shot manner, such as "target speed of an agent should be 8m/s" or "minimize the power usage of an agent".

To this end, we present a novel LLM-based Skill Diffusion (LDuS) framework, designed to facilitate zero-shot adaptation to unseen contexts by generating skill trajectories that are controllable through loss-guided diffusion. Specifically, we devise a hierarchical skill learning structure in which a diffusion model is employed as a skill planner with sequential in-painting. This in-painting method sequentially substitutes consecutive state-action pairs with their originals and learns on the remaining parts, thus allowing for robust trajectory generation by conditioning the trajectory on past experiences. For evaluation, LDuS provides an interface where a context specified in language is translated into a loss function to guide the generation process of the skill diffusion planner. The generation process is then continuously refined through an iterative process using an LLM as a self-critic, ensuring that the generating skill trajectory aligns with the given context. As such, our framework stands apart from existing language-conditioned skill imitation approaches, as it enables the zero-shot adaptation of skill-based policies to various contexts that extend beyond the training dataset.

The contributions of our work are summarized as follows.

- We present the LDuS framework to address a novel challenge of zero-shot policy adaptation to unseen contexts specified in language.

- We develop a hierarchical skill learning structure that adapts the skill diffusion planner with sequential in-painting, enabling robust skill trajectory generation.

- We devise an interface that utilizes LLMs to translate a context into a loss function, which is then used to control the generation process of the skill diffusion planner. This is further validated via iterative refinement to adequately align with the given context.

- We experimentally show that LDuS achieves superior performance in zero-shot adaptation to a wide range of contexts including different specification levels, multi-modality, and varied temporal conditions for robotic manipulation tasks.

## 2 Related work

### 2.1 Language-conditioned skill learning

In the domain of sequential decision-making, several researches have explored techniques for learning language-conditioned skills [1, 2, 3, 6, 7, 8]. LISA [2] employs hierarchical skill learning to achieve

a language-conditioned policy through discretized skill codes. Recently, LCD [3], PlayFusion [6], and SkillDU [8] commonly adopt diffusion models as a language-conditioned policy to address high-dimensional vision inputs or to leverage play datasets collected by human. While these approaches primarily concentrate on learning by direct supervision from language-annotated datasets, our LDuS aims at adapting to unseen contexts that convey varying environment conditions or different user requirements beyond the datasets.

Meanwhile, several studies have harnessed the code generation capabilities of LLMs to ground language instructions to actionable skills [9, 10, 11, 12, 13, 14]. For example, in Kinematic-LLM [14], an LLM is prompted with in-context samples to generate the waypoints for robotic manipulation with pre-defined code primitives. The performance of this in-context learning often depends on the quality and relevance of selected samples, leading to limited generalization to unseen contexts. Unlike this waypoint-based high-level planning with pre-defined code primitives, our LDuS adapts the code generation capabilities of LLMs with iterative refinement to enable the fine-grained control of trajectories, particularly suited for adjusting the generation process of diffusion models.

## 2.2 Guided control for diffusion models

Diffusion models have shown promising results in various areas including computer vision [15, 16, 17, 18], offline reinforcement learning (RL) [19, 20, 21], and long-horizon planning [22, 23, 24, 25]. The strong generation ability of diffusion models leads to robust adaptability at deployment. In classifier guidance [26], diffusion models are controlled at test-time, thereby supporting the generation of images belonging to specific classes. This controlled adaptation concept has been further investigated for text-driven image generation [27, 28] and noisy inverse problems [29, 30]. Recently, LGD [4] presents a loss-guided diffusion mechanism, by which diffusion models can be controlled via differentiable loss functions without additional training. To facilitate the query-compliant scene generation, CTG [5] leverages LLMs as a loss function generator for user queries.

In the RL domain, such guidance schemes for controlling diffusion-based policies have been investigated with pre-trained dynamics models [31] and value functions [22, 23]. Yet, these schemes rarely accommodate language-specified contexts. Our LDuS is the first to integrate the reasoning capabilities of LLMs and the controlled generation capabilities of diffusion models, thus enabling zero-shot policy adaptation to language-specified contexts in the domain of sequential decision-making.

# 3 Preliminaries

## 3.1 Problem formulation

**Contextual Markov Decision Process (MDP).** We consider a task as a contextual MDP [32, 3] $(\mathcal{C}, S, A, \mathcal{G}, P^c, r^c, \gamma, \rho_0)$ where $c \in \mathcal{C}$ is a context space, $s \in S$ is a state space, $a \in A$ is an action space, $g \in \mathcal{G}$ is a goal space, $P^c : \mathcal{C} \times S \times A \times S \to [0,1]$ is a transition probability conditioned on the context and the goal, $r^c : \mathcal{C} \times S \times A \times \mathcal{G} \to \mathbb{R}$ is a reward function conditioned on the context, $\gamma \in [0,1]$ is a discount factor, and $\rho_0 : S \to [0,1]$ is an initial state distribution. Here, we consider a goal to be specified in language, denoted as $g_l$, such as "open the drawer" or "close the door". Moreover, we assume a context is provided in language [3], denoted as $c_l$, which can affect either the reward function or the transition probability.

**Policy adaptation to contexts.** We assume access to a dataset $\mathcal{D} = \{\tau_i\}_{i \leq N}$, where each trajectory $\tau_i$ is represented as a sequence of state and action pairs with a goal $\{(s_t, a_t, g_l)\}_{t \leq T}$ for a $T$-length episode without any contextual information.

We consider task evaluation scenarios with a context $c_l$ which conveys environmental conditions or user requirements, along with a goal $g_l$. Then, our objective is to develop a policy adaptation framework $\phi(g_l, c_l)$ that maps both goal $g_l$ and context $c_l$ to a policy $\pi_c$ maximizing the return of context-conditioned rewards.

$$\phi^* = \underset{\phi}{\mathrm{argmax}} \underset{\substack{g_l \sim \mathcal{G}, c_l \sim \mathcal{C}, \\ \pi_c \sim \phi(g_l, c_l), a_t \sim \pi_c(\cdot | s_t, g_t)}}{\mathbb{E}} \left[ \sum_{t=0}^{T-1} r^{c_l}(s_t, a_t, g_l) \right] \tag{1}$$

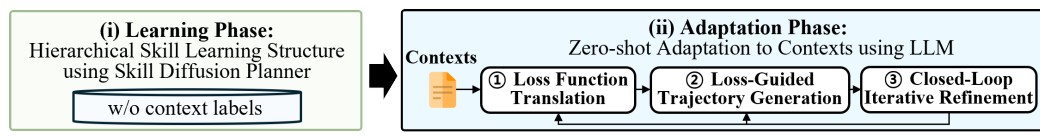

Figure 2: Concept of LDuS with skill learning and adaptation phases

## 3.2 Diffusion probabilistic models

Diffusion models have been explored for task planning [22, 23, 24, 33], offline RL [19, 20, 21], and language-conditioned skill learning [3, 6, 8]. In [22], a diffusion model for planning was introduced, which generates $h$-length state and action sequences, denoted as $x$ in a two-dimensional array.

$$x = \begin{bmatrix} s_0 & s_1 & \cdots & s_h \\ a_0 & a_1 & \cdots & a_h \end{bmatrix} \qquad (2)$$

In particular, the diffusion model $\epsilon(x^k, k)$ based on the U-net architecture [34] predicts a noise $\eta \sim \mathcal{N}(0, I)$ given a noise-corrupted trajectory $x^k$ as an input [15, 16], and it is optimized through the following loss.

$$\min_{\epsilon} \mathbb{E}_{x \sim \mathcal{D}, k \sim [1,K]}[||\epsilon(x^k, k) - \eta||_2^2] \qquad (3)$$

Here, $k \in [1, K]$ is a denoising step, and $x^k = \sqrt{\alpha^k}x + \sqrt{1 - \alpha^k}\eta$ is generated by adding a Gaussian noise $\eta$ to the original trajectory $x$ with a variance schedule parameter $\alpha^k$. At sampling, the diffusion model generates trajectory $x$ from a random noisy input $x^K \sim \mathcal{N}(0, I)$ by sequentially denoising it,

$$x^{k-1} = \frac{1}{\sqrt{\alpha^k}}\left(x^k - \frac{1-\alpha^k}{\sqrt{1-\bar{\alpha}^k}}\epsilon(x^k, k)\right) + \sigma^k \eta, \qquad (4)$$

where $\sigma^k$ is a parameter for a variance schedule.

## 4 Our approach

### 4.1 Overall framework

To address the zero-shot adaptation to various contexts, we develop the LDuS framework comprising two phases: (i) skill learning via the skill diffusion planner, and (ii) policy adaptation to unseen contexts via LLM-guided diffusion, as illustrated in Figure 2. (i) In the learning phase, we establish a hierarchical structure in which skills are learned, conditioned on a goal. This structure includes a skill encoder, a skill prior, and a skill diffusion planner that generates skill trajectories. These components are learned on the dataset that contains only goals, without any contextual information. Additionally, we employ a sequential in-painting technique when training the skill diffusion planner to enhance robustness in skill trajectory generation. (ii) In the adaptation phase, we guide the generation process of the skill diffusion planner upon a language-specified context, by harnessing the code generation capabilities of LLMs. The generated code serves as a loss function that guides the skill diffusion planner to generate skill trajectories at every denoising step. This facilitates alignment between the skill trajectories and the given context. Furthermore, LDuS employs an iterative refinement process, in which generated skill trajectories are repeatedly validated in a closed-loop manner to achieve robust alignment with the context.

### 4.2 Skill learning via diffusion planner

To facilitate skill learning using a diffusion model, we adopt a variational autoencoder (VAE) [35] architecture with three components: a skill encoder $q(z|s_{t:t+h}, a_{t:t+h})$, a skill prior $p_l(z|s_t, g_l)$, and a skill diffusion planner $\epsilon(x^k, k, z)$. Given an $h$-length skill trajectory $x = \{s_t, a_t\}_{t \leq h}$, the skill encoder $q$ predicts a skill embedding $z$, then the skill diffusion planner reconstructs the $h$-length skill trajectory $x$ based on $z$. To optimize both the skill encoder and skill diffusion planner, we employ a conditional VAE objective that combines a diffusion reconstruction term in (3) and a prior regularization term such as

$$\min_{q,\epsilon} \mathbb{E}_{x \sim \mathcal{D}, z \sim q, k \sim [1,K]} \left[ ||\epsilon(x^k, k, z) - \eta||_2^2 \right] + \beta D_{\text{KL}}\left(q(z|s_{t:t+h}, a_{t:t+h}), p(z)\right) \qquad (5)$$

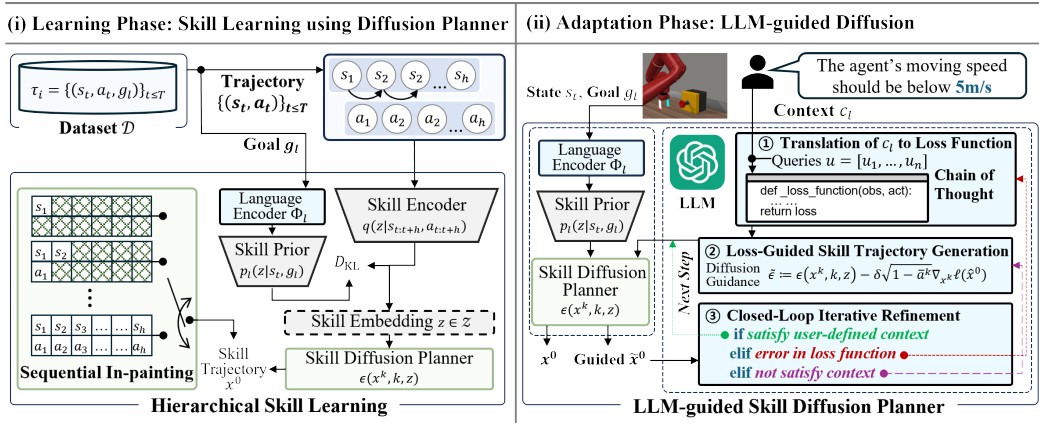

Figure 3: LDuS framework: In (i), given the dataset annotated with the goals, skills are learned through the hierarchical structure employing a skill diffusion planner with sequential in-painting techniques. In (ii), the context specified in language is translated into a loss function, which is then used to guide the generation process of the skill diffusion planner. This process is further validated with closed-loop iterative refinement to better align skill trajectories with the given context.

where $p(z)$ is a unit Gaussian $\mathcal{N}(0, I)$, $D_{\text{KL}}$ is the Kullback-Leibler (KL) divergence, and $\beta$ is a weight for regularization [36]. To establish a versatile skill embedding space encapsulating common skills across multiple tasks, we use the skill encoder without goal conditions. In addition, to learn skills conditioned on the goal, the skill prior $p_l$, which is conditioned on state $s_t$ and goal $g_l$, is tailored to align with the output of the skill encoder. To handle the goal provided in language, we also use a pre-trained language encoder $\Phi_l$ such as CLIP [37] that produces language embeddings of $g_l$ for the skill prior. Then, the skill prior is jointly trained with the skill encoder and skill diffusion planner by minimizing the distance with the skill encoder, i.e.,

$$\min_{p_l} D_{\text{KL}}\left(q(z|s_{t:t+h}, a_{t:t+h}), p_l(z|s_t, \Phi_l(g_l))\right). \tag{6}$$

Using the skill prior, we predict an appropriate skill embedding at deployment based on the current state and specified goal.

**Sequential in-painting.** The in-painting technique is adopted in [22] to address goal-conditioned problems, in which a diffusion model is conditioned by replacing the last state of a generating trajectory $x^k$ with the goal state. We adapt this in-painting to train the skill diffusion planner with a sequential replacement mechanism, where a sequence of $m \sim [1, h]$ states and actions of $x^k$ is substituted with the corresponding original state-action pairs to learn the remaining portion of $x^k$, as illustrated in the bottom left side of Figure 3. Then, for evaluation, unlike conventional methods that constrain the diffusion model only with the current state [22, 23], our approach constrains the skill diffusion planner using the previously encountered $m$ states. This in-painting method enables the diffusion planner to generate more robust and contextually aligned skill trajectories, as described in Section 5.3.

## 4.3 Policy adaptation via LLM-guided diffusion

In general, it is challenging for a model to directly acquire the zero-shot adaptation ability for various contexts, particularly when the training dataset has limited coverage on those contexts. To tackle this challenge, we harness the controlled generation capabilities of diffusion models [26] along with the code-generation capabilities of LLMs [5].

In LDuS, for zero-shot adaptation to various contexts, we employ three procedures: translation of a given context to its corresponding loss function, loss-guided trajectory generation, and closed-loop iterative refinement, as illustrated in the right side of Figure 3. Initially, the LLM is tasked with translating the context into a loss function. This loss function then guides the skill diffusion planner in its trajectory generation process. To ensure the accuracy and relevance of generated skill trajectories, the process is validated by the LLM in a closed-loop manner, querying the LLM to verify whether the generated trajectories meet the specifications of the given context.

---

**Algorithm 1** Policy adaptation via LLM-guided diffusion

---

1: **Inputs**: skill prior $p_l(z|s_t, g_l)$, skill diffusion planner $\epsilon(x^k, k, z)$, goal $g_l$, context $c_l$, LLM $\Phi_{\text{LLM}}$, guidance weight $\delta$
2: Obtain loss function $\ell(\hat{x}^0)$ using $\Phi_{\text{LLM}}$ through (7)
3: **for** every environment step $t$ **do**
4:     $z \sim p_l(z|s_t, g_l)$
5:     **while** not validate **do**
6:         Sample trajectory $x^0$ using $\epsilon(x^k, k, z)$ without guidance through (4)
7:         Sample guided trajectory $\tilde{x}^0$ using $\ell(\hat{x}^0)$ and $\epsilon(x^k, k, z)$ through (8)
8:         Validate whether satisfy the context via LLM as $\Phi_{\text{LLM}}(x^0, \tilde{x}^0, g_l, c_l)$
9:     Execute $a_m$ in $x^0$ to the environment

---

**Translation of context to loss function.** For converting the context into a loss function in code, we employ chain-of-thought prompting [38] with a pre-defined list of queries $\boldsymbol{u} = [u_1, .., u_n]$. The queries are designed to capture the specifications of the agent and the desired format of loss functions. Then, the queries are sequentially prompted to the LLM $\Phi_{\text{LLM}}$ in conjunction with a goal $g_l$ and a context $c_l$, i.e.,

$$\{y_j | y_j = \Phi_{\text{LLM}}(g_l, c_l, \{u_i, y_i\}_{i<j})\} \tag{7}$$

where $\{u_i, y_i\}_{i<j}$ represents a set of prompted queries $u_i$ and their respective responses $y_i$ from the LLM. The final response $y_n$ is then used as the loss function to guide the skill diffusion planner.

**Loss-guided skill trajectory generation.** Similar to prior work [29, 4], we implement the loss-guided skill trajectory generation, where the guidance is computed as the gradient of the loss function with respect to $x^k$ at each denoising step $k$.

$$\tilde{\epsilon} := \epsilon(x^k, k, z) - \delta\sqrt{1 - \bar{\alpha}^k}\nabla_{x^k}\ell(\hat{x}^0) \tag{8}$$

Here, $\hat{x}^0 = \frac{1}{\sqrt{\bar{\alpha}^k}}(x^k + (1 - \bar{\alpha}^k)\epsilon(x^k, k, z))$ is an approximation of $x^0$ given $x^k$ [29], and $\delta$ is a hyperparameter to modulate the strength of the guidance.

**Closed-loop iterative refinement.** In the closed-loop iterative refinement, we employ the LLM as a self-critic to evaluate both the loss functions and generated trajectories. Specifically, we prompt the LLM with unguided trajectory $x^0$, guided trajectory $\tilde{x}^0$, goal $g_l$, and context $c_l$, i.e., $\Phi_{\text{LLM}}(x^0, \tilde{x}^0, g_l, c_l)$. The LLM then checks for errors. If errors are detected in the loss function, it is regenerated. If there is a mismatch between the trajectory and the context, the frequency of the guidance application is increased. This ensures continuous improvement in the accuracy and relevance of trajectories generated by the skill diffusion planner. The process of zero-shot policy adaptation is summarized in Algorithm 1.

## 5 Experiments

### 5.1 Experiment Settings

**Datasets.** We use the MetaWorld benchmark [39], specifically with 10 different robot manipulation goals. We also utilize long-horizon goals from the multi-stage MetaWorld, where each goal comprises a sequence of short-horizon manipulation sub-goals. For data collection, we emulate rule-based expert policies. For each goal, we collect 60 trajectories, varying the speed of the agent as well as the position and weight of the objects being manipulated.

**Contexts.** We use two context groups: (i) **language context** where the context is solely specified in language, (ii) **multi-modal context** where additional information is provided through image input to assist in resolving the given context. These contexts are used to direct the agent with instructions such as moving below or above a specific speed, adjusting its speed faster or slower along a specified axis, or exerting more or less force on heavy objects. The context details are in Appendix A.4.

**Evaluation metrics.** We use two metrics to assess the zero-shot performance of LDuS and the baselines. **Success Rate (SR)** quantifies the percentage of goals or sub-goals that are successfully completed. **Context Reward (CR)** evaluates the average reward achieved based on how effectively the models satisfy the given context.

Table 1: Zero-shot performance: The baselines and LDuS are trained on 10 different manipulation goals for MetaWorld and 3 different long-horizon goals for multi-stage MetaWorld. For each manipulation goal, we use $2 \sim 5$ different contexts. The success rate (SR) and context rewards (CR) are measured in $95\%$ confidence interval. Each is evaluated with 5 random seeds for language contexts and 3 random seeds for multi-modal contexts. The highest performance is highlighted in bold.

(a) Performance in MetaWorld

| Method | Without context | Language context | | Multi-modal context | |
| --- | --- | --- | --- | --- | --- |
| | SR (%) | CR | SR (%) | CR | SR (%) |
| LangDT | $38.15 \pm 7.42\%$ | $27.18 \pm 2.65$ | $33.16 \pm 7.19\%$ | $0.00 \pm 0.00$ | $0.00 \pm 0.00\%$ |
| LISA | $11.11 \pm 6.00\%$ | $17.58 \pm 6.95$ | $10.32 \pm 4.13\%$ | $0.00 \pm 0.00$ | $0.00 \pm 0.00\%$ |
| LCD | $52.98 \pm 8.90\%$ | $37.64 \pm 6.10$ | $50.98 \pm 8.44\%$ | $4.58 \pm 3.00$ | $11.11 \pm 4.00\%$ |
| Diffuser | $92.16 \pm 3.37\%$ | $57.66 \pm 6.20$ | $86.90 \pm 3.71\%$ | $0.18 \pm 0.63$ | $0.00 \pm 0.00\%$ |
| LCD + Guidance | - | $42.04 \pm 4.70$ | $49.95 \pm 8.35\%$ | $22.13 \pm 0.87$ | $0.00 \pm 0.00\%$ |
| Diffuser + Guidance | - | $69.77 \pm 4.23$ | $76.42 \pm 5.26\%$ | $33.01 \pm 0.52$ | $1.38 \pm 1.01\%$ |
| LDuS (ours) | $\mathbf{97.00 \pm 0.73\%}$ | $\mathbf{87.36 \pm 3.40}$ | $\mathbf{94.60 \pm 1.70}\%$ | $\mathbf{63.52 \pm 4.28}$ | $\mathbf{93.05 \pm 2.29}\%$ |

(b) Performance in multi-stage MetaWorld

| Method | Without context | Language context | | Multi-modal context | |
| --- | --- | --- | --- | --- | --- |
| | SR (%) | CR | SR (%) | CR | SR (%) |
| LangDT | $5.83 \pm 2.32\%$ | $3.82 \pm 1.71$ | $4.86 \pm 1.91\%$ | $0.00 \pm 0.00$ | $4.20 \pm 9.30\%$ |
| LISA | $1.17 \pm 1.25\%$ | $30.56 \pm 0.00$ | $0.93 \pm 0.68\%$ | $0.00 \pm 0.00$ | $0.00 \pm 0.00\%$ |
| LCD | $35.27 \pm 3.23\%$ | $32.11 \pm 7.02$ | $35.84 \pm 2.51\%$ | $34.65 \pm 13.69$ | $20.85 \pm 7.94\%$ |
| Diffuser | $35.55 \pm 3.77\%$ | $45.49 \pm 4.23$ | $40.66 \pm 3.29\%$ | $25.00 \pm 13.79$ | $18.75 \pm 10.36\%$ |
| LCD + Guidance | - | $39.37 \pm 4.81$ | $35.28 \pm 2.30\%$ | $53.69 \pm 7.06$ | $29.20 \pm 6.89\%$ |
| Diffuser + Guidance | - | $45.89 \pm 3.20$ | $36.67 \pm 2.81\%$ | $46.73 \pm 7.57$ | $18.75 \pm 7.17\%$ |
| LDuS (ours) | $\mathbf{81.95 \pm 2.42\%}$ | $\mathbf{81.99 \pm 2.86}$ | $\mathbf{84.03 \pm 3.67}\%$ | $\mathbf{82.03 \pm 4.20}$ | $\mathbf{60.45 \pm 8.46}\%$ |

**Baselines.** For comparison, we use several language-conditioned imitation and planning methods. 1) **LangDT** [40] is an imitation learning method that utilizes a language-conditioned decision transformer, 2) **LISA** [2] is a hierarchical skill imitation framework that learns discredited skill codes conditioned on language instructions, 3) **LCD** [3] is a hierarchical planning framework that reconstructs state sequences using a diffusion model conditioned on a language input, 4) **Diffuser** [22] is a task planning framework based on diffusion models.

For LLMs, we use GPT-3.5 [41] which is capable of generating loss functions in the form of executable code. Moreover, for multi-modal contexts, we use GPT-4 [42]. Since these baselines rarely account for zero-shot adaptation to contexts, we adopt the same diffusion guidance used in LDuS for those (i.e., Diffuser and LCD) employing diffusion models. In the cases where the diffusion model is guided by a hand-designed loss function, which is considered optimal, we specify such baselines with the additional label of **Guidance**.

## 5.2 Main results

**Zero-shot performance.** Table 1 shows the performance of LDuS and the baselines (LangDT, Diffuser, LISA, LCD) across three different context input cases (without context, language context, multi-modal context) in MetaWorld and multi-stage MetaWorld. As in Table 1(a), LDuS consistently yields the best SR and CR in MetaWorld, outperforming the most competitive baseline Diffuser+Guidance by $18.2\%$ higher in SR and $78.7\%$ higher in CR for the cases of language contexts. For multi-stage MetaWorld, in Table 1(b), LDuS demonstrates superior performance with $41.7\%$ higher in SR and $75.5\%$ higher in CR at average, compared to Diffuser+Guidance.

In these experiments, LangDT and LISA exhibit the lowest performance, even for the cases without contexts. This is attributed to the multi-modality in the dataset, which tends to hinder the learning of policies built with multi-layer perceptrons or transformers. In contrast, the baselines employing diffusion models, such as Diffuser and LCD, show improved performance. However, none of these

Table 2: Performance w.r.t various context types

| Method | Precise context | | Abstract context | | Temporal context | |
|--------|-----------------|------|-----------------|------|-----------------|------|
| | CR | SR (%) | CR | SR (%) | CR | SR (%) |
| LCD + Guidance | $30.35 \pm 6.55$ | $47.96 \pm 8.68\%$ | $50.18 \pm 4.17$ | $48.71 \pm 9.04\%$ | $25.04 \pm 9.02$ | $33.80 \pm 3.18\%$ |
| Diffuser + Guidance | $65.05 \pm 4.79$ | $70.19 \pm 4.87\%$ | $57.33 \pm 2.05$ | $50.58 \pm 4.68\%$ | $49.17 \pm 6.59$ | $38.89 \pm 0.68\%$ |
| LDuS (ours) | $89.77 \pm 3.99$ | $97.72 \pm 1.08\%$ | $75.52 \pm 3.86$ | $86.98 \pm 3.00\%$ | $79.40 \pm 3.32$ | $84.26 \pm 3.79\%$ |

baselines achieve robust comparable performance to LDuS for the cases involving language or multi-modal contexts. This limitation arises because the baselines are primarily designed to handle goal descriptions, which is the sole form of language annotation in the dataset. Consequently, they lack the capability to accommodate various contexts that convey environmental conditions or user requirements. Diffuser and LCD, when used with guidance, exhibit slightly improved CR, as the hand-designed optimal loss function can provide context-aligned guidance for trajectory generation. However, in some cases, SR slightly decreases for both the baselines and LDuS when guidance is applied. This decrease occurs because the gradient-based loss-guidance could generate unexpected trajectories unless the tuning of the guidance weight $\delta$ was carefully managed. Overall, LDuS outperforms the baselines in CR by employing the iterative refinement that ensures context alignment, as well as in SR by employing the sequential in-painting that allows for robust trajectory generation. While the contexts used in Table 1 are mainly related to speed, we provide additional experiments on different types of contexts, such as energy constraints and spatial limitations, in Appendix C.2.

**Various context types.** In Table 2, we evaluate the performance across several context types such as precise, abstract, and temporal contexts, while all given contexts are specified in language. Specifically, precise contexts include detailed user requirements, such as a specific target speed, e.g., "the agent speed should move between 5m/s and 6m/s." In contrast, abstract contexts lack specific details. For instance, if the user wants the agent to increase its speed, the abstract context could be phrased as "I am very busy; the agent needs to hurry." Temporal contexts are dynamic and vary over time, which are particularly relevant in long-horizon goals. As shown, the results indicate that LDuS significantly enhances CR, with an increase of $43.7\%$ at average compared to Diffuser+Guidance.

**Comparing with waypoint generation.** In addition to the learning-based baselines compared previously, we compare our approach with Kinematic-LLM [14] by which waypoints for pre-defined skill primitives are generated through an LLM with in-context samples. To implement Kinematic-LLM, we define basic skill primitives such as move, push, and pull, and use the same samples for prompting, which are used for LDuS. As shown in Table 3, Kinematic-LLM shows comparable performance in SR for MetaWorld, but lags in multi-stage MetaWorld. This is attributed to the increased complexity of planning with LLMs for long-horizon goals. Regarding CR, Kinematic-LLM consistently demonstrates lower performance compared to LDuS. This result stems from a lack of versatile samples and skill primitives that are necessary to effectively adapt to various contexts.

Table 3: Comparison with waypoint generation method

| Method | MetaWorld | | Multi-stage MetaWorld | |
|--------|-----------|------|----------------------|------|
| | CR | SR (%) | CR | SR (%) |
| Kinematic-LLM | $48.83 \pm 3.80$ | $95.24 \pm 1.53\%$ | $51.19 \pm 0.93$ | $61.82 \pm 3.05\%$ |
| LDuS (ours) | $90.38 \pm 3.14$ | $95.36 \pm 1.57\%$ | $75.99 \pm 3.81$ | $84.03 \pm 3.67\%$ |

**Inference Time.** In Table 4, we present the average inference time (in milliseconds) required per timestep for LDuS and the baselines. The measurements are conducted on a system equipped with an Intel(R) Core(TM) i9-10980XE CPU and an NVIDIA RTX A6000 GPU, and we use GPT-3.5 for the LLM. As LDuS requires both diffusion sampling time and LLM inference, we measure these component separately, denoted as "Diffusion" and "LLM" in the parenthesis. Diffuser and LCD exhibit the shortest inference time, as these baselines do not require loss guided sampling or LLM inference. When considering only the diffusion sampling time excluding LLM inference, LDuS demonstrates an inference time comparable to the baselines that use the loss guidance. However, the full inference time of LDuS is longer due to its LLM-based code generation and iterative refinement

process. This overhead can be mitigated by using a smaller language model, which can be obtained by distilling the essential knowledge required for LDuS.

Table 4: Inference time required per timestep

| Method | Diffuser | LCD | Diffuser+Guidance | LCD+Guidance | LDuS |
|---|---|---|---|---|---|
| Inference Time | 55ms | 56ms | 102ms | 100ms | 108ms(Diffusion) + 55ms(LLM) |

**Skill trajectory coverage.** Figure 4 illustrates the t-SNE embeddings of $h$-length trajectories presented in the dataset (yellow-colored dots) and skill trajectories generated by LDuS with guidance (blue-colored dots) for MetaWorld. We collect successful skill trajectories from LDuS, using different contexts and varying the guidance weight $\delta$ ranging from $0.05$ to $0.4$. This weight setting regulates the strength of gradient application, as described in (8). As observed, the embeddings are expanded via LDuS, specifying that LDuS is capable of rendering novel skill trajectories, which

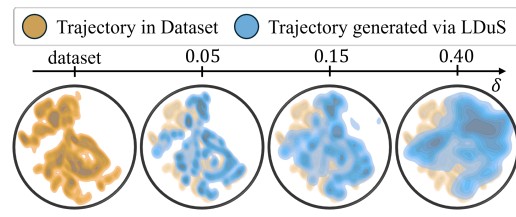

Figure 4: Skill trajectory coverage

are not presented in the dataset but necessary to adapt to different contexts. This demonstrates the versatility of LDuS that arises from the generation capabilities of the diffusion model.

## 5.3 Ablation study

**LLM-guided skill diffusion planner.** Table 5 shows the impact of our LLM-guided skill diffusion planner. In this ablation study with MetaWorld, we evaluate two variants of LDuS, one without loss guidance and the other without iterative refinement. LDuS achieves improved CR of $37.8\%$ over the variant without the loss guidance for language contexts. It also achieves improved CR of $46.8\%$ on average over the variant without iterative refinement. These results underscore the effectiveness of LDuS with loss guidance and iterative refinement for zero-shot adaptation.

Table 5: Ablation on LLM-guided skill diffusion planner

| Method | Language context | | Multi-modal context | |
|---|---|---|---|---|
| | CR | SR (%) | CR | SR (%) |
| LDuS | $90.38 \pm 3.14$ | $95.36 \pm 1.57\%$ | $63.52 \pm 4.28$ | $93.05 \pm 2.29\%$ |
| $-$ Loss Guidance | $65.58 \pm 5.02$ | $95.50 \pm 0.96\%$ | $0.58 \pm 0.10$ | $0.00 \pm 0.00\%$ |
| $-$ Iterative Refinement | $85.74 \pm 4.28$ | $94.34 \pm 1.17\%$ | $33.75 \pm 0.48$ | $4.17 \pm 1.36\%$ |

**Sequential in-painting.** Figure 5 shows the effect of our sequential in-painting technique. With multi-stage MetaWorld, the hatched bars denote the performance of LDuS and Diffuser without sequential in-painting, while the solid-colored bars indicate that of these models with sequential in-painting. SR and CR are significantly improved for both LDuS and Diffuser, when the in-painting technique is applied. This is attributed to conditioning on the past $m$ experiences, which function as long-term memory. This memory feature enhances performance, particularly for long-horizon goals like multi-stage MetaWorld. Furthermore, the performance enhancement is more pronounced for LDuS, as it benefits from learning common skills from the dataset containing trajectories of multiple goals.

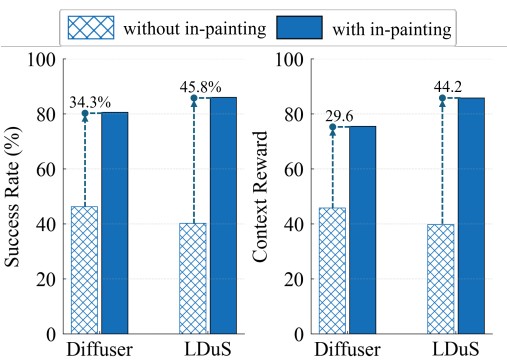

Figure 5: Ablation on sequential in-painting

# 6 Conclusion and limitations

In this work, we presented the LDuS framework for zero-shot skill-based policy adaptation to contexts specified in language. The framework employs a hierarchical structure for skill learning, in which the skill encoder learns task-agnostic skill abstractions and the skill diffusion planner generates various skill trajectories. The skill diffusion planner is enhanced with sequential in-painting, thus enabling context-aligned trajectory generation for the skills. At test-time, given a specific context describing environmental conditions or user requirements, LDuS directly influences the generation process of the skill diffusion planner, allowing for skill-based policies to adapt to the context. This zero-shot adaptation is achieved by a combination of LLM-based loss function generation and iterative refinement, along with the controllable structure of the skill diffusion planner. LDuS stands apart from other language-conditioned approaches, which are limited to certain variations of the instructions present in the dataset and generalize insufficiently to a range of unseen contexts.

**Limitations.** LDuS has several limitations which direct us to future work. One limitation is related to the inference time, as LDuS relies on iterative LLM inferences for refinement. This issue could be mitigated by distilling only essential knowledge, such as code generation and verification capabilities of an LLM, into a smaller language model. Furthermore, since LDuS relies on the LLM for multiple components, including code generation and iterative refinement, another limitation is its robustness, which can be affected by the variability in the LLM's performance and the design of the prompts.

## Acknowledgements

This work was supported by Institute of Information & communications Technology Planning & Evaluation (IITP) grant funded by the Korea government (MSIT) (No. RS-2022-II220043 (2022-0-00043) Adaptive Personality for Intelligent Agents, RS-2022-II221045 (2022-0-01045) Self-directed multi-modal Intelligence for solving unknown, open domain problems, RS-2019-II190421, Artificial Intelligence Graduate School (Sungkyunkwan University)), by ICT Creative Consilience Program through the Institute of Information & communications Technology Planning & Evaluation (IITP) grant funded by the Korea government (MSIT) (No. Rs-2020-II201821), by the National Research Foundation of Korea (NRF) grant funded by MSIT (No. RS-2023-00213118), by BK21 FOUR Project (No. S-2024-0580-000), and by Samsung electronics.

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

# A Benchmark environments

In this section, we provide detailed information about our environment settings, dataset collection strategy, and context configurations used for zero-shot evaluation.

## A.1 MetaWorld

For evaluation, we use MetaWorld [39], where an agent is tasked with manipulating an object to achieve a given goal. We use 10 different manipulating goals in MetaWorld including press button, open window, close window, open door, open drawer, close drawer, pick place cube, insert peg, push cube, open faucet. The left side of Figure 6 illustrates several manipulating goals in MetaWorld. For zero-shot evaluation, we utilize all manipulating goals for language contexts and select two manipulating goals for multi-modal contexts.

## A.2 Multi-stage MetaWorld

To evaluate long-horizon scenarios, we modify MetaWorld to configure the long-horizon goals [43]. Each goal in the multi-stage Metaworld consists of four existing MetaWorld sub-goals including slide puck, close drawer, push button and insert peg. The agent is then tasked to complete these sub-goals in a specified order. We use 3 different long-horizon goals in this multi-stage MetaWorld, each with a unique sub-task completion sequence. The goal description for multi-stage MetaWorld is formed by concatenating the descriptions of the sub-goals in the specified order, such as "close drawer and insert peg and push button and slide puck". The right side of Figure 6 illustrates an example of long-horizon goal used in the multi-stage MetaWorld. For zero-shot evaluation, we utilize all three long-horizon goals for language contexts and one long-horizon goal for multi-modal context.

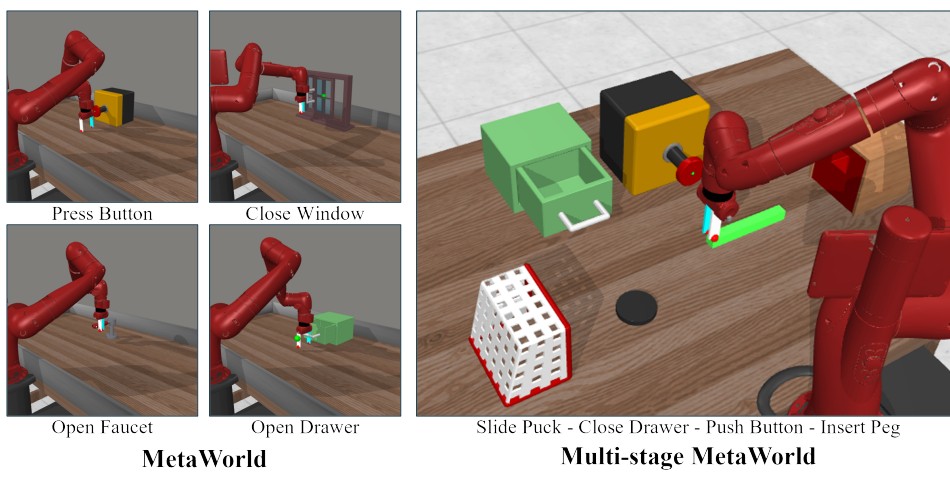

Figure 6: Visualization of Benchmark Environments

## A.3 Dataset collection

To generate training datasets, we implement rule-based policies for each goal in Metaworld and multi-stage MetaWorld. These rule-based policies are configured using 6 different skill primitives: move along the x-axis, move along the y-axis, move along the z-axis, push, pull, and grab. Each skill primitive operates based on Proportional-Integral-Derivative (PID) control. To emulate diverse expert behaviors, we vary the speed of the agent, as well as the position and weight of the manipulation objects. For each goal, we collect 60 successful trajectories.

## A.4 Context configuration

We configure various contexts specified in natural language to evaluate the zero-shot adaptability of LDuS and the baselines. These contexts are defined based on different specification levels (abstract

and precise), multi-modality, and varied temporal conditions. To assess whether LDuS and the baselines align with the given context, we manually design context-specific reward functions for each scenario.

For **precise contexts**, a concrete user requirement is given with a numerical value. We define four different precise contexts as follows:

- The agent should move at a *speed slower* than $x$ m/s
- The agent should move at a *speed greater* than $x$ m/s
- The agent should move at a *speed faster* than $x$ m/s but *slower* than $x$ m/s
- The agent should move at *speed along the y-axis a minimum* of $x$ m/s.

Here, $x$ is filled with different numerical values, depending on the specific goal and the context. To configure the zero-shot adaptation settings, we test the model with speeds different from those used in the training dataset.

For **abstract contexts**, a user requirement is conveyed with some degree of ambiguity. We define two different abstract contexts as follows:

- I want to be relaxed, but the agent is *too distracting*.
- I'm very busy, the agent should *hurry up*.

For **multimodal context**, the agent is required to exert greater force only when in context with the manipulating object, aiming to minimize energy consumption of the end executor. In this scenario, the agent utilizes image input to determine whether it is in context with the manipulating object.

- The agent should exert a greater speed of $x$ m/s only *when in contact with the object*.

For **temporally varying context**, the context changes over time to one of the precise contexts.

## B   Implementation details

In this section, we provide the implementation details of LDuS and other baselines, along with the hyperparameter settings used for training. All experiments are conduced on a system equipped with an Intel(R) Core(TM) i9-10980XE CPU and an NVIDIA RTX A6000 GPU.

### B.1   LangDT

We implement LangDT based on Decision Transformer [40]. LangDT consists of a single decision transformer, built on GPT-2 [44], conditioned by the goal specified in language. Since LangDT is trained solely by conditioning on goals, which are the only language annotations in the dataset, it struggles with unseen contexts. Therefore, for zero-shot evaluation, we condition the model solely on the given goal. The hyperparameter settings for LangDT are summarized in Table 6 Note that *short* refers to MetaWorld, while *long* refers to multi-stage MetaWorld.

Table 6: Hyperparameter settings for LangDT

| Hyperparameter | Value |
|---|---|
| Total timesteps | 1e6 |
| Batch size | 8 (per goal) |
| Learning rate | 3e-5 |
| Embedding size | 128 |
| Hidden size | 512 |
| Number of heads | 4 |
| Number of layers | 2 |
| Planning horizon | 8 (*short*), 16 (*long*) |

## B.2 LISA

We implement LISA [2] using the open source project [2]. LISA consists of a skill predictor and a policy, both implemented with causal transformers. The skill predictor generate discrete skill codes given a goal and a current state. Then, the policy generates a $h$-length skill based on the predicted skill code. Similar to LangDT, LCD is evaluated by conditioning the model solely on the goal. The hyperparameter settings for LISA are summarized in Table 7.

Table 7: Hyperparameter settings for LISA

| Hyperparameter | Value |
|---|---|
| Epoch | 2500 |
| Batch size | 8 (per goal) |
| Policy learning rate | 1e-4 |
| Skill predictor learning rate | 1e-4 |
| Language model learning rate | 1e-6 |
| Skill horizon | 10 |
| Number of skills | 20 |

## B.3 LCD

We implement LCD [3] using the open source projects Jax [3] and Haiku [4]. LCD consists of a diffusion model, which generates a state trajectory given a current state and a goal, and an inverse dynamics mode, which reconstructs actions given a pair of states. For zero-shot adaptation, we implement LCD controlled with a loss function, denoted as LCD+Guidance in the main manuscript. The loss functions are hand-designed for each context, and loss guidance is applied at each denoising timestep of the diffusion model, following the same approach as in the loss-guided trajectory generation process of LDuS. The hyperparameter settings for LCD are summarized in Table 8.

Table 8: Hyperparameter settings for LCD

| Hyperparameter | Value |
|---|---|
| Total timesteps | 1e6 |
| Batch size | 8 (per goal) |
| Learning rate | 3e-5 |
| Denoising timesteps | 20 |
| Variance scheduler | cosine |
| Planning horizon | 8 (*short*), 16 (*goal*) |

## B.4 Diffuser

We implement Diffuser [22] using the open source projects Jax and Haiku. Diffuser is butile with a diffusion model that generates a trajectory given a current state and a goal. Similar to LCD, we implement Diffuser controlled with a manually designed loss function, which is denoted as Diffuser+Guidance in the main manuscript. The hyperparameter settings for Diffuser are summarized in Table 9.

## B.5 Kinematic-LLM

We implement Kinematic-LLM [14] using GPT-3.5 [41]. Kinematic-LLM consists of a kinematic knowledge parser and a kinematic-aware planner. The kinematic knowledge parser generates an XML file that describes the current state of the environment, including details about the kinematic properties

---

[2]https://github.com/Div99/LISA
[3]https://github.com/google/jax
[4]https://github.com/google-deepmind/dm-haiku

Table 9: Hyperparameter settings for Diffuser

| Hyperparameter | Value |
|---|---|
| Total timesteps | 1e6 |
| Batch size | 8 (per goal) |
| Learning rate | 3e-5 |
| Skill embedding size | 128 |
| Denoising timesteps | 20 |
| Variance scheduler | cosine |
| Planning horizon | 8 (*short*), 16 (*long*) |

of both the manipulating object and the agent. Following this, the kinematic-aware planner generates an abstract manipulation sequence along with 3D manipulation waypoints. The abstract manipulation sequence provides the description of the procedure, while the 3D manipulation waypoints give the numerical values of the skill primitives in a structured format. The LLM is prompted with the generated XML file containing current kinematic information, a goal description, the available skill primitive, and sufficient in-context samples. For skill primitives, we use the 6 different skills defined for rule-based expert policy emulation, as described in Section A.3. To evaluate zero-shot performance on unseen contexts, the language context is concatenated to the goal description. Below, we provide the prompt used for Kinematic-LLM.

```
# N Number of In-context Samples

## In-context Sample 1

### Goal Description:
    Faucet Open

### Object Kinematic Knowledge:
    <hand pos="0.006 0.400 0.195"/>
    <joint pos="0.007 0.839 0.124" type="hinge" range="-1.57 1.57"
    axis="0 0 1"/>
    <faucet pos="0.007 0.689 0.124"/>

### Guidance:
    First generate abstract sequence, further convert into waypoints

### Actions for execution:
    [MOVE_X] target_x (grab)
    [MOVE_Y] target_y (grab)
    [MOVE_Z] target_z (grab)
    [GRIP]
    [PUSH] target_x target_y target_z (grab)
    [PULL] target_x target_y target_z (grab)

### Abstract Manipulation Sequence:
    1  Align with faucet's y position (0.689)
    2. Go to faucet's x position (0.007)
    3. Go smaller x value than faucet's x position (-0.093)
    4. Align with faucet's z position (0.124)
    5. Now push the faucet toward front left direction (0.207, 0.889,
    0.124)

### 3D Manipulation Waypoints:
    [MOVE_Y] 0.683
    [MOVE_X] 0.007
    [MOVE_X] -0.093
    [MOVE_Z] 0.124
    [PUSH] 0.207 0.889 0.124

## In-context Sample 2
    ...
```

```
## In-context Sample N
    ...

--------------------------------------------------------------------------
# It is your turn to generate the waypoints for the specified task.
    You should fill in Abstract Manipulation Sequence and 3D
    Manipulation Waypoints sections.

### Goal Description:
    Faucet Open

### Object Kinematic Knowledge:
    <hand pos="0.006 0.400 0.195"/>
    <joint pos="-0.013 0.840 0.124" type="hinge" range="-1.57 1.57"
    axis="0 0 1"/>
    <faucet pos="-0.013 0.690 0.124"/>

### Guidance:
    First generate abstract sequence, further convert into waypoints

### Actions for execution:
    [MOVE_X] target_x (grab)
    [MOVE_Y] target_y (grab)
    [MOVE_Z] target_z (grab)
    [GRIP]
    [PUSH] target_x target_y target_z (grab)
    [PULL] target_x target_y target_z (grab)

### Abstract Manipulation Sequence:

### 3D Manipulation Waypoints:
```

Listing 1: The prompt used for Kinematic-LLM

## B.6   LDuS

We implement LDuS using the open source projects Jax and Haiku. LDuS consists of a skill encoder based on an LSTM, a skill prior based on MLPs, and a skill diffusion planner. At test-time, the skill prior is used to predict the skill embedding given a goal and a current state. Then, the skill diffusion planner generates skill trajectory based on the skill embedding. To embed goals specified in language, we utilize CLIP [37]. The hyperparameter settings for LDuS are summarized in Table 10.

Table 10: Hyperparameter settings for LDuS

| Hyperparameter | Value |
| --- | --- |
| Total timesteps | 1e6 |
| Batch size | 8 (per goal) |
| Learning rate | 3e-5 |
| Input embedding size | 128 |
| Skill embedding size | 64 |
| Hidden size | 128 |
| Denoising timesteps | 20 |
| Variance scheduler | cosine |
| Planing horizon | 8 (*short*), 16 (*long*) |

**Translation of context to loss function.** We manually design queries that describe the specifications of the agent and the desired format of the loss function. These queries are used in chain-of-thought (CoT) prompting to guide the LLM in generating a loss function form a given context. We utilize GPT-3.5 [41]. Below, we present the prompt used to translate contexts to loss functions for LDuS.

```
Query 1: "The shape of the action sequence of the agent is (B, H, 4)
    where B is the batch size, H is the number of sequences, and 4
    represents the (x, y, z, grab on/off) in corresponding agent
    coordinate of the agent. Explain the configuration the agent's
    action"

Query 2: "The agent should satisfy the given user requirement. The
    user requirement is given as follows: [context]. Note that the
    speed of the agent is determined by the L2 norm of the actions. In
     order to satisfy the user requirement, what should be considered?
    "

Query 3: "Now, generate a loss function that guides the generating
    trajectory to satisfy the given user requirement. Generated Python
     based loss function should follow the following format: 'def
    _loss_fn(x, obs_dim): act = [x:,:,obs_dim] return loss', where act
     is a numpy array representing the action sequences."
```

Listing 2: The prompt used for generating loss function

Then, the generated loss function is used to guide the generation process of our skill diffusion planner. Below, we present an example of code generated by the LLM.

```
# Goal Description
Faucet Open

# Context:
The agent should move at a speed faster than 0.38 but slower than
    0.40.

# Generated Loss Function:
def _loss_fn(x, obs_dim):
    act = x[:,:,obs_dim]
    speed = np.linalg.norm(act, axis=-1)
    min_speed = 0.38
    max_speed = 0.40
    loss = np.maximum(speed - max_speed, 0) + jnp.maximum(min_speed -
    speed, 0)
    return np.mean(loss)
```

Listing 3: An example of code generated by the LLM

**Loss-guided skill trajectory generation.** We implement the loss-guided skill trajectory generation based on previous studies [4, 29]. In practice, instead of predicting noise from a noise-corrupted trajectory $x^k$, our skill diffusion planner reconstructs the original trajectory $x^0$ to ensure robust generation. Thus, we do not need to explicitly calculate $\hat{x}^0$ from $x^k$, and the loss-guidance is directly applied to the output of the skill diffusion planner. The frequency of guidance applications, denoted as $n$, is determined by an iterative refinement procedure, where $n$ is gradually increasedfrom 1 to a pre-defined maximum value until the generated trajectory aligns with the given context. The procedure of loss-guided skill trajectory generation is summarized in Algorithm 2.

**Closed-loop iterative refinement.** We use an LLM to detect errors in the loss function and identify any mismatch between the generated trajectory and the given context. Below, we present the prompt used to regenerate the loss function when LDuS detects such errorsn.

```
Query : "The previously generated code has an error. Regenerate the
    Python code. Note that the user requirement was [context]."
```

Listing 4: The prompt used to detect error in the loss function

Below, we present the prompt use to detect a mismatch. If a mismatch is found between the generated trajectory with guidance and the context, the frequency of the guidance application is increased. This validation process is conducted every $h$ steps using the LLM.

**Algorithm 2** Loss-guided skill trajectory generation

1: **Inputs**: skill diffusion planner $\epsilon(x^k, k, z)$, loss function $\ell$, total denoising timestep $K$, guidance weight $\delta$, frequency for guidance application $n$
2: $x^K \sim \mathcal{N}(0, I)$
3: **for** $k = K - 1, ..., 0$ **do**
4:     $x^k_{s_0:s_m} \leftarrow s_{0:m}$
5:     $\tilde{\epsilon} \leftarrow \epsilon(x^k, k, z)$
6:     **for** $i = 1, ...n,$ **do**
7:         $\tilde{\epsilon} \leftarrow \tilde{\epsilon} - \delta \nabla_{x^k} \ell(\hat{x}^0 = \tilde{\epsilon})$
8:     $\eta \sim \mathcal{N}(0, I)$
9:     $x^{k-1} \leftarrow \frac{\sqrt{\alpha^k}(1-\bar{\alpha}^{k-1})}{1-\bar{\alpha}^k}x^k + \frac{\sqrt{\alpha^{k-1}}\beta^k}{1-\bar{\alpha}^k}\tilde{\epsilon} + \sigma^k \eta$
10: **return** $x^0$

```
Query 1: "The originally generated trajectory is [unguided trajectory]
    and modified trajectory is [trajectory with guidance]. Did the
    modified action satisfy the given user requirement? The user's
    requirement is [context]."

Query 2: "According to the answer above, did the modified action meet
    user's requirement? Answer with 'Yes' or 'No'".
```

Listing 5: The prompt used to detect a mismatch between generated trajectory and the context

**Multi-modal context.** For a multi-modal context, LDuS needs to determine whether the agent is in contact with the manipulating object from an image input. We utilize GPT-4 [42], as it can concurrently process both image and text inputs. If the LLM detects that the agent is in contact with the object, the loss-guidance is applied. Initially, we generate a caption describing the image using the LLM. Then, we prompt the LLM to assess whether the agent is in contact with the object based on the caption. To further enhance the reasoning capability of the LLM, we prompt the LLM with the previous image and the corresponding answer to better understand the context of the current image. Below, we present the prompt used for a multi-modal context.

```
Caption : "In the image, the red robot arm is attempting to close the
    green drawer with white drawer handle. The robot's arm has a
    cylindrical blue and white tool attached to its end, which is
    called as an end effector. The background is an indoor setting
    with a wooden floor and a gray wall."

Query : "[caption]. [current image] Given the image, is the robot
    contacting on the handle of the drawer? [previous image] The
    asnwer for the previous image was 'No'."
```

Listing 6: The prompt used for multi-modal context

## C  Additional experiments

### C.1  Detailed experiment results

Table 11 and 12 shows the detailed experiment results in MetaWorld and multi-stage MetaWorld, respectively, under the conditions where no contexts are provided. As shown, LDuS consistently achieves the best performance in SR for all goals. The low performance of LangDT and LISA is attributed to the multi-modal nature of the dataset, which hinders the learning of policies built with MLPs or transformers. LCD and Diffuser mitigate this issue to some extent by employing the diffusion models, but their performance is still lower compared to LDuS. This performance gap is more pronounced in multi-stage MetaWorld. This is because LDuS leverages as sequential in-painting method, which contributes to more robust trajectory generation.

Table 13 and 14 show the detailed experiment results in MetaWorld under precise language contexts, as described in A.4. In these table, we report the results of LCD and Diffusion with guidance only,

Table 11: Performance without context in MetaWorld

| Method | Press Button | Open Window | Close Window | Open Door | Open Drawer |
|---|---|---|---|---|---|
| LangDT | $36.7 \pm 5.5\%$ | $68.3 \pm 9.7\%$ | $83.3 \pm 8.5\%$ | $6.7 \pm 3.1\%$ | $23.3 \pm 8.6\%$ |
| LISA | $27.8 \pm 9.2\%$ | $19.4 \pm 12.4\%$ | $30.6 \pm 14.8\%$ | $8.3 \pm 4.7\%$ | $5.5 \pm 6.0\%$ |
| LCD | $31.6 \pm 12.0\%$ | $60.0 \pm 11.0\%$ | $93.3 \pm 3.1\%$ | $20.0 \pm 8.8\%$ | $36.7 \pm 2.4\%$ |
| Diffuser | $96.7 \pm 2.4\%$ | $100.0 \pm 0.0\%$ | $100.0 \pm 0.0\%$ | $100.0 \pm 0.0\%$ | $93.3 \pm 4.9\%$ |
| LDuS (ours) | $100.0 \pm 0.0\%$ | $100.0 \pm 0.0\%$ | $100.0 \pm 0.0\%$ | $80.0 \pm 0.0\%$ | $100.0 \pm 0.0\%$ |

| Method | Close Drawer | Pick Place Cube | Insert Peg | Push Cube | Open Faucet |
|---|---|---|---|---|---|
| LangDT | $40.0 \pm 5.7\%$ | $45.0 \pm 15.1\%$ | $3.3 \pm 2.4\%$ | $41.6 \pm 12.5\%$ | $33.3 \pm 3.1\%$ |
| LISA | $8.3 \pm 4.7\%$ | $0.0 \pm 0.0\%$ | $0.0 \pm 0.0\%$ | $0.0 \pm 0.0\%$ | $11.1 \pm 8.1\%$ |
| LCD | $90.0 \pm 3.3\%$ | $41.6 \pm 14.2\%$ | $33.3 \pm 14.9\%$ | $53.3 \pm 14.9\%$ | $70.0 \pm 4.3\%$ |
| Diffuser | $80.0 \pm 0.0\%$ | $83.3 \pm 8.1\%$ | $86.6 \pm 9.7\%$ | $91.7 \pm 5.3\%$ | $90.0 \pm 3.3\%$ |
| LDuS (ours) | $100.0 \pm 0.0\%$ | $96.7 \pm 2.4\%$ | $93.3 \pm 4.9\%$ | $100.0 \pm 0.0\%$ | $100.0 \pm 0.0\%$ |

Table 12: Performance without context in multi-stage MetaWorld: We abbreviate each goal by using initials of its words (e.g. "Close Drawer" is CD, and "Push Button" is PB)

| Method | CD-SP-IP-PB | SP-CD-PB-IP | PB-CD-SP-IP |
|---|---|---|---|
| LangDT | $12.5 \pm 3.9\%$ | $4.2 \pm 2.5\%$ | $0.8 \pm 0.6\%$ |
| LISA | $0.7 \pm 0.8\%$ | $1.4 \pm 1.5\%$ | $1.4 \pm 1.5\%$ |
| LCD | $40.0 \pm 3.7\%$ | $43.3 \pm 3.4\%$ | $22.5 \pm 2.6\%$ |
| Diffuser | $46.7 \pm 1.4\%$ | $36.7 \pm 5.1\%$ | $23.3 \pm 4.7\%$ |
| LDuS (ours) | $78.3 \pm 1.4\%$ | $75.0 \pm 3.5\%$ | $92.5 \pm 2.4\%$ |

as they are the most comparable baselines. As shown, LDuS consistently delivers robust zero-shot performance in both CR and SR across all configurations. In multi-stage MetaWorld, the performance fluctuates in the baselines, due to their limited ability to generate plans for long-horizon goals.

Table 13: Zero-shot performance with language context in MetaWorld

| Method | Context1 | | Context2 | |
|---|---|---|---|---|
| | CR | SR (%) | CR | SR (%) |
| LCD + Guidance | $30.6 \pm 8.5$ | $55.7 \pm 8.8\%$ | $34.7 \pm 4.5$ | $53.7 \pm 8.4\%$ |
| Diffuser + Guidance | $64.0 \pm 3.0$ | $57.7 \pm 8.4\%$ | $63.5 \pm 6.6$ | $91.5 \pm 2.6\%$ |
| LDuS (ours) | $82.1 \pm 5.9$ | $94.3 \pm 1.7\%$ | $91.0 \pm 1.7$ | $96.3 \pm 1.0\%$ |

| Method | Context3 | | Context4 | |
|---|---|---|---|---|
| | CR | SR (%) | CR | SR (%) |
| LCD + Guidance | $49.1 \pm 3.5$ | $53.8 \pm 8.1\%$ | $53.8 \pm 2.3$ | $45.7 \pm 9.3\%$ |
| Diffuser + Guidance | $73.8 \pm 3.7$ | $85.1 \pm 6.4\%$ | $77.8 \pm 3.6$ | $86.7 \pm 4.7\%$ |
| LDuS (ours) | $86.7 \pm 3.2$ | $96.7 \pm 0.8\%$ | $89.6 \pm 2.0$ | $90.0 \pm 3.6\%$ |

Table 14: Zero-shot performance with language context in multi-stage MetaWorld

| Method | Context1 | | Context2 | |
|---|---|---|---|---|
| | CR | SR (%) | CR | SR (%) |
| LCD + Guidance | $58.2 \pm 1.4$ | $35.3 \pm 2.7\%$ | $20.5 \pm 8.2$ | $35.3 \pm 1.9\%$ |
| Diffuser + Guidance | $48.1 \pm 3.4$ | $10.6 \pm 1.8\%$ | $43.2 \pm 3.4$ | $62.8 \pm 3.8\%$ |
| LDuS (ours) | $75.1 \pm 3.2$ | $87.0 \pm 3.2\%$ | $88.9 \pm 2.5$ | $81.0 \pm 3.6\%$ |

## C.2 Performance on different context types

In Table 15, we present additional experiments involving two different context types: energy constraints and spatial limitations. We conduct these experiment on a single task in MetaWorld. For the energy context, the agent aims to minimize its energy consumption by reducing acceleration or deacceleration [45]. For the spatial context, the agent is tasked with stying within a specified spatial boundary without crossing it. As shown, LDuS outperforms the baselines in CR and SR, demonstrating its scalability cross diverse context types.

However, for complex tasks such as dexterous control, interactions with the environment [46] or AP functions [47] are necessary. This is because LLMs are not inherently grounded in these complex environments, making it challenging for them to directly generate directly loss functions. We believe that incorporating such approaches will enable LDuS to accommodate more complex tasks, which we plan to explore as part of our future research directions.

Table 15: Zero-shot performance with energy and spatial context in MetaWorld

| Method | Energy Context | | Spatial Context | |
|---|---|---|---|---|
| | CR | SR (%) | CR | SR (%) |
| LCD + Guidance | $56.3 \pm 0.3$ | $66.7 \pm 30.8\%$ | $29.4 \pm 38.8$ | $33.3 \pm 0.0\%$ |
| Diffuser + Guidance | $62.3 \pm 0.3$ | $66.7 \pm 30.8\%$ | $75.4 \pm 37.5$ | $100.0 \pm 0.0\%$ |
| LDuS (ours) | $88.8 \pm 0.1$ | $100.0 \pm 0.0\%$ | $87.0 \pm 16.6$ | $100.0 \pm 0.0\%$ |

