# OpenReview forum: "LLM-based Skill Diffusion for Zero-shot Policy Adaptation"
_NeurIPS.cc/2024/Conference — NeurIPS 2024 poster_

### Official Review · Reviewer_UqKP · 2024-07-02

**Soundness:** 2
**Presentation:** 3
**Contribution:** 2
**Rating:** 5
**Confidence:** 4

**Summary:**

This paper presents a novel framework called LLM-based Skill Diffusion (LDuS), designed to enable zero-shot skill-based policy adaptation to various contexts specified in natural language. It leverages LLMs to guide a skill diffusion model, allowing for the generation of adaptable skill trajectories. The framework uses a hierarchical skill learning structure and a loss-guided diffusion process with sequential in-painting techniques. The authors demonstrate the effectiveness of LDuS in adapting to diverse contexts through experiments on robotic manipulation tasks, showing improved performance compared to other methods.

**Strengths:**

* The approach is interesting and tackles an important problem. The method appears to be novel to my knowledge on the topic.

* The method is tested on various challenging tasks and achieves superior performance compared to several baselines, demonstrating its effectiveness.

**Weaknesses:**

### Major comments:
* The combination of a diffusion model and a VAE to learn the skills from a dataset of trajectories is not well justified. In particular, Equation (5), which is the loss function for learning the skills, informally replaces the reconstruction loss in ELBO of VAE with the noise prediction loss of a diffusion model. It is unclear if this is mathematically and conceptually correct. If this equation is adapted from a paper, it should be cited. Otherwise, more justification for this combination is needed. The authors are encouraged to look into works that combine these two models, such as:

Kingma, Diederik, Tim Salimans, Ben Poole, and Jonathan Ho. "Variational diffusion models." Advances in neural information processing systems 34 (2021): 21696-21707.


* The seeding procedure is unclear and possibly incorrect. In the caption of Table 1, the authors state that the methods are "evaluated" on 3 seeds. This suggests that they trained on a single seed and evaluated on 3 seeds, which is not the correct use of seeding. The proper approach is to train and evaluate for each seed. Additionally, more seeds should be used to ensure robustness. While computation costs are a concern, using at least 5 seeds, and preferably 10, is recommended.
* The contribution of the paper does not appear to be significant. It combines multiple components into a framework. In some cases these combinations are not well-justified either (e.g., VAE+diffusion mentioned above). While I understand the results are promising, I have concerns about the significance of the contributions.


### Minor comments:
* Notation fix: In Equation 1, the action sampling needs to be added to the subscript of the expectation: $a_t \sim \pi_c(\cdot | s_t, g_t)$.

* Given that VAE is one of the main components of the method, its paper should be cited:

Kingma, Diederik P., and Max Welling. "Auto-encoding variational bayes." arXiv preprint arXiv:1312.6114 (2013).

* The cross-reference to Table 3 in the related works section is out of place. Table 3 presents empirical results, and referring to it during the literature review breaks the coherence of the paper and the order of cross-references.

* A bit nit picky, but the abbreviation "LDuS" for "LLM-based skill diffusion" is unclear and should be clarified.

**Questions:**

1. What is the justification for Equation (5)?
2. Please explain the seeding procedure.

**Limitations:**

The authors briefly discuss one limitation of their work in line 320. However, more discussion is needed given the multiple components and heavy reliance on LLMs for tasks such as encoding the goal with CLIP or encoding the context with GPT. Additionally, potentially negative societal impacts are not discussed.

---

> ### Author Rebuttal · Authors · 2024-08-06
>
> ## [Weakness 1 & Question 1] Mathematical Justification for VAE+Diffusion
> We appreciate the reviewers for addressing concerns about the mathematical justification LDuS. We provide the mathematical justification for Equation (5), where the reconstruction loss in the ELBO is replaced with the diffusion loss. Due to length constraints in the rebuttal text, **the complete proof is included in the PDF of the global rebuttal**.
> We start from the ELBO formulation for VAEs [1],
>
> $$\mathop{\mathbb{E}}\_{x_T \sim q(x_T|x,z), z \sim q(z|x)} [\log p(x)] \geq \underbrace{\mathop{\mathbb{E}}\_{x_T \sim q(x_T|x,z), z \sim q(z|x)} \left[\log p(x|z)\right]}\_{\text{reconstruction loss}} - \underbrace{D_{\text{KL}}(q(z|x)||p(z))}\_{\text{regularization loss}}$$
> Then, the reconstruction error term satisfies the following,
>
> $$\mathop{\mathbb{E}}\_{x_T \sim q(x_T|x,z), z \sim q(z|x)} [\log p(x|z)]  \geq \mathop{\mathbb{E}}\_{x_T \sim q(x_T|x,z), z \sim q(z|x)} \left[\log p(x_T)\right] + \underbrace{\mathop{\mathbb{E}}\_{x_T \sim q(x_T|x,z), z \sim q(z|x)} \left[ \log \frac{p(x|x_T,z)}{q(x_T|x,z)}\right]}\_{\text{diffusion loss}}$$
> where $T$ is the total denoising timestep. Thus, the reconstruction term can be seamlessly replaced with the diffusion loss. Moreover, on how the diffusion loss term is optimized with the noise reconstruction term, please refer to the original DDPM paper [2]. We will include this theoretical analysis in the final version.
>
> Moreover, our LDuS framework is structurally different from the approach presented in the "Variation Diffusion Models (VDM)" [3] paper. In VDM, a diffusion model itself functions as a VAE. In contrast, LDuS utilizes an encoder-decoder structure, where an encoder generates skill embeddings from trajectories, and a diffusion-based skill planner produces skill trajectories conditioned on these embeddings. Moreover, using VAE structure to learn skill embeddings and decode skills is a common practice in skill-based RL [4,5].
>
> [1] Kingma, Diederik P., and Max Welling. "Auto-encoding variational bayes." arXiv 2013
>
> [2] Ho, Jonathan, et al. "Denoising diffusion probabilistic models." NeurIPS 2020
>
> [3] Kingma, Diederik, et al. "Variational diffusion models." NeurIPS 2021
>
> [4] Pertsch, Karl, et al. "Accelerating reinforcement learning with learned skill priors." CoRL 2020
>
> [5] Hakhamaneshi, Kourosh, et al. "Hierarchical few-shot imitation with skill transition models." ICLR 2022
>
> ----
>
> ## [Weakness 2 & Question 2] Random Seed
> We used 3 different seeds to **train** and evaluate our model. To avoid misunderstanding, we will revise the caption in Table 1.
>
> In the below table, we report the performance of LDuS and the baselines on 5 random seeds. As shown, LDuS consistently outperforms the baselines, demonstrating the robustness of our experiments. In the final version, we will conduct experiments using 5 or more random seeds for all cases.
>
> |Method|Without Context (SR)|Language Context (CR)|Language Context (SR)|
> |-|-|-|-|
> |LCD+Guidance|52.9\%|42.0|49.9\%|
> |Diffuser+Guidance|92.2\%|69.8|76.4\%|
> |LDuS|97.0\%|87.4|94.6\%|
>
> ----
> ## [Weakness 3] Contribution of LDuS
> We believe our work addresses a novel practical problem where the agent adapts to unseen language-specified contexts in a zero-shot manner, even when trained without a context-labeled dataset. Previous works on language-conditioned skill-learning [1,2] primarily focus on learning through direct supervision from language-annotated datasets. However, obtaining datasets annotated with a broad range of contexts is impractical in real-world scenarios due to the open-ended nature of language. Furthermore, in the realm of RL, while diffusion-based policies have been explored primarily for imitating given datasets [3,4], language-guided test-time control for zero-shot adaptation has not been thoroughly investigated. To the best of our knowledge, our LDuS is the first to integrate LLMs with diffusion models for zero-shot adaptation to language-specified contexts in the domain of sequential decision-making.
>
> [1] Garg, Divyansh, et al. "LISA: Learning interpretable skill abstractions from language." NeurIPS 2022
>
> [2] Zhang, Edwin, et al. "Language control diffusion: Efficiently scaling through space, time, and tasks." ICLR 2024
>
> [3] Wang, Zhendong, et al. "Diffusion policies as an expressive policy class for offline reinforcement learning." ICLR 2024
>
> [4] Chen, Chang, et al. "Simple hierarchical planning with diffusion." ICLR 2024
>
> ----
> ## [Weakness 4,5,6] Minor Comments
> * Subscript in Equation 1: We will add the subscript for the Equation 1.
> * Citation: We will cite the paper "Auto-encoding variational bayes" in Section 3.2.
> * Cross-Reference: We will remove this reference to Table3 in the related works, to enhance the coherence of our paper.
> ----
> ## [Weakness 7] Abbreviation of LDuS
> As the reviewer mentioned, the abbreviation "LDuS" is unclear. Initially, we intended to name our model 'LSDu', based on the order in "LLM-based Skill Diffusion".  However, to avoid the association with the acronym 'LSD', which is commonly known for the drug, we rearranged the letters, resulting in the final model name 'LDuS'.
>
> ----
> ## [Limitations] Reliance on LLMs
> The LDuS framework relies on LLMs for its performance; however, it maintains robustness through an iterative refinement process, with its effectiveness detailed in Table 4. For goal description, using RoboCLIP [1], a vision language model trained on robot manipulation tasks, would improve the representation of the goal descriptions. Additionally, the use of LLMs to control the generation process of diffusion models raises concerns about potential negative societal impacts, such as the possibility of malicious prompts being injected and used for harmful purposes. We will discuss these concerns in the limitation section and provide potential negative social impacts.
>
> [1] Sontakke, Sumedh, et al. "Roboclip: One demonstration is enough to learn robot policies." NeurIPS 2024

---

> > ### Comment · Reviewer_UqKP · 2024-08-08
> >
> > I thank the authors for their reply. Most of my concerns were addressed and I have increased my score.

---

> ### Author Response · Authors · 2024-08-09
>
> We are very satisfied that your concerns have been addressed. We truly appreciate your valuable feedback, as well as your decision to increase the score.
>
> We fully agree with your perspective that a mathematical justification for the VAE+Diffusion model is required. As mentioned in our rebuttal, the reconstruction loss in the VAE can be seamlessly replaced by the diffusion loss. We will include this justification in the final version of our paper.
>
> We once again deeply thank you for your thoughtful feedback on our paper.

---

### Official Review · Reviewer_MnsC · 2024-07-07

**Soundness:** 3
**Presentation:** 3
**Contribution:** 3
**Rating:** 5
**Confidence:** 3

**Summary:**

This paper presents LDuS, a framework that adapts skill diffusion models to unseen contexts in a zero-shot manner. The proposed method first couples VAE with a diffusion planner for hierarchical skill learning. To perform zero-shot policy adaptation to the language-specified context, LDuS translates contexts to loss functions using LLM and leverages loss-guided diffusion for controllable trajectory generation. Furthermore, LDuS adopts LLM to iteratively evaluate and improve the alignment between the generated trajectory and the given context. The method is extensively evaluated in two zero-shot settings of MetaWorld, and achieves superior performance in both success rate and context reward compared to the baselines.

**Strengths:**

- LDuS innovatively integrates the reasoning ability of LLM and loss-guided diffusion for policy adaptation and effectively improves the zero-shot performance given unseen contexts.
- The authors further demonstrate the robustness of LDuS in different context types, and provide more insights on LDuS capabilities through the results of waypoint generation and trajectory coverage.
- The ablation study highlights the importance of several design choices.
- The illustration of motivation and methodology is sound and clear, and the paper is well-written and organized.

**Weaknesses:**

- Tested contexts seem to be mainly limited in speed specifications.
- More seeds should be tested in experiments to draw more convincing and robust conclusions.
- The application of loss guidance may harm the model's capability of completing the goal/task (i.e. success rate).

**Questions:**

- Would sequential in-painting increase the training cost (e.g. training time) of the learning phase?
- How is the context reward defined specifically for each context?
- In line 228, how is the loss function manually designed for the "Guidance" setting?
- How many times of refinement are usually needed before a generated trajectory is considered contextually aligned? Would that be too time-consuming for long-horizon tasks?
- How sensitive is LDuS to loss guidance strength?

**Limitations:**

The authors adequately discussed the limitations of the proposed method.

---

> ### Author Rebuttal · Authors · 2024-08-06
>
> ## [Weakness 1] Scope of Context
> As the scope of context is limited in our experiments, we conduct additional experiments on two different types of user requirements including energy and spatial. In the energy context, the agent aims to minimize its energy consumption by reducing its acceleration or deceleration [1]. In the spatial context, the agent is tasked to not cross a specified spatial boundary. In the table below, we conduct tests on a single task in MetaWorld. As shown, LDuS outperforms the baselines in context reward (CR), demonstrating its scalability on diverse context types.
>
> |Method|Energy Context (CR)|Energy Context (SR)|Spatial Context (CR)|Spatial Context (SR)|
> |---|---|---|---|---|
> |LCD+Guidance|56.3|66.7\%|29.4|33.3\%|
> |Diffuser + Guidance|62.3|66.7\%|75.4|100.0\%|
> LDuS|88.8|100\%|87.0|100.0\%|
>
> [1] Soori, Mohsen, et al. "Optimization of energy consumption in industrial robots, a review." Cognitive Robotics 2023
>
> ----
>
> ## [Weakness 2] Random Seed
> In the below table, we conduct the additional experiments using 5 different seeds for LDuS and the baselines (Diffuser, LCD).  As shown, LDuS consistently outperforms the baselines, demonstrating the robustness of our experiments. Moreover, we conduct 120 different combinations of task and context to draw the result for language context, in our tables. In the final version, we will conduct additional experiments using 5 or more random seeds for all cases. Moreover, we conduct 120 different combinations of task and context to draw the result for language context, in our tables.
>
> |Method|Without Context (SR)|Language Context (CR)|Language Context (SR)|
> |---|---|---|---|
> |LCD+Guidance|52.9\%|42.0|49.9\%|
> |Diffuser+Guidance|92.2\%|69.8|76.4\%|
> |LDuS|97.0\%|87.4|94.6\%|
>
> ----
>
> ## [Weakness 3] Loss Guidance
> While there is potential for loss guidance to impair the model's ability to complete tasks, a similar gradient-based diffusion control approach has been extensively studied across various domains such as vision [1,2,3,4], motion generation [5,6], traffic scene generation [7], and reinforcement learning [8,9,10]. These studies consistently demonstrate its robustness in diverse applications. Moreover, we have shown that our LDuS framework achieves robust performance using loss guidance, further validating its effectiveness.
>
> [1] Ho, Jonathan, et al. "Classifier-free diffusion guidance." NeurIPS 2022
>
> [2] Kwon, Gihyun, et al. "Improving Diffusion-based Image Translation using Asymmetric Gradient Guidance." ICML 2023
>
> [3] Dinh, Anh-Dung, et al. "PixelAsParam: A gradient view on diffusion sampling with guidance." ICML 2023
>
> [4] Liu, Xihui, et al. "More control for free! image synthesis with semantic diffusion guidance." WACV 2023
>
> [5] Song, Jiaming, et al. "Loss-guided diffusion models for plug-and-play controllable generation." ICML 2023
>
> [6] Karunratanakul, Korrawe, et al. "Guided motion diffusion for controllable human motion synthesis." CVPR 2023
>
> [7] Zhong, Ziyuan, et al. "Language-guided traffic simulation via scene-level diffusion." CoRL 2023
>
> [8] Janner, Michael, et al. "Planning with diffusion for flexible behavior synthesis." ICML 2022
>
> [9] Ni, Fei, et al. "Metadiffuser: Diffusion model as conditional planner for offline meta-rl." ICML 2023
>
> [10] Liang, Zhixuan, et al. "Adaptdiffuser: Diffusion models as adaptive self-evolving planners." ICML 2023
>
> ----
>
> ## [Question 1] Training Cost
> As the sequential in-painting requires optimization with respect to $m \sim [1,h]$, the training time is increased for LDuS at the learning phase. However, since our objective is to enable zero-shot policy adaptation, efficiency at the training time is not our primary concern. Nonetheless, we ensured that all baselines received a sufficient amount of training time to achieve convergence.
>
> ----
>
> ## [Question 2] Definition of Context Reward
> For **precise contexts**, where specific user requirements such as speed are quantified numerically, we calculate the difference between the specified and actual values. In cases of **abstract contexts**, where user requirements are expressed ambiguously (without explicit numerical values), we assess context reward by comparing the actual speed with the average speed in the dataset. For **multi-modal context**, where the agent should follow the user requirement only when it makes contact with the manipulating object, we compute context reward by calculating the difference between the specified and actual value when contact is made. Additionally, since the agent should not apply loss guidance without making contact, we impose penalties when the agent attempts to provide loss guidance while not in contact with the object.
>
> ----
>
> ## [Question 3] Loss Function Design for "Guidance"
> The loss functions are manually designed by the domain experts for each context in the "Guidance" setting. To ensure the optimality of the loss functions, we conducted multiple tests and revisions to effectively reflect the given contexts.
>
> ----
>
> ## [Question 4] Number of Iterative Refinement
> The refinement process is required between 0 to 4 depending on the context and the tasks. Moreover, our iterative refinements are executed for every skill horizon. For long-horizon tasks, we set the skill horizon at 16 steps, compared to 8 steps for short-horizon tasks. Consequently, the time required for iterative refinement is not significantly increased for long-horizon tasks.
>
> ----
>
> ## [Question 5] Sensitivity to Guidance Strength
> LDuS is less sensitive to the loss guidance strength, as it employs iterative refinement to control the frequency of guidance application. A small guidance weight is preferable, because it allows for more detailed control. However, a small guidance weight may necessitate a more iterative refinement process to meet the specified context, presenting a tradeoff between detailed control and increased inference time.

---

> ### Comment · Reviewer_MnsC · 2024-08-09
> **Rebuttal Acknowledgement**
>
> I want to thank the authors for their efforts in adding experiments and detailed responses to my questions. I suggest that the authors also include the standard deviation in the final paper for better comparison. Most of my concerns have been addressed, I will maintain my positive rating towards the paper.

---

> > ### Author Response · Authors · 2024-08-10
> >
> > We deeply appreciate your positive rating of our paper and are very satisfied that your concerns have been addressed. Due to the text limit on the rebuttal, we did not include the standard deviation; however, it will be included in our final paper.
> >
> > Regarding to the scope of the context, we will include experiments both 'spatial' and 'energy' contexts in our final paper. Additionally, we will address the several questions raised by the reviewer, such as the definition of context reward, the number of iterative refinement required, and sensitivity to guidance strength in our final paper.
> >
> > We thanks for the reviewer once again for your thoughtful response.

---

### Official Review · Reviewer_261B · 2024-07-12

**Soundness:** 3
**Presentation:** 3
**Contribution:** 3
**Rating:** 7
**Confidence:** 3

**Summary:**

LDuS is a diffusion-based approach for offline skill learning that adopts several advances to improve performance in goal-driven settings. The main contribution is the adoption of LLM-based guidance, allowing to comply with contextual information/conditions, while achieving the given goal.

**Strengths:**

The work presents advances that improve skill diffusion and allow to comply with additional contextual information. The strengths of the work are:
* **Novelty**: the work presents two mechanisms, sequential in-painting and LLM-guided skill diffusion, which are somehow incremental but contribute in a novel way to the field. Given that these topics are of particular interest, I consider the novelty introduced significant.
* **Presentation**: the work is clearly presented, well-structured and with high-quality figures and tables
* **Evaluation**: the evaluation of the work is extensive, in terms of tasks, including various context types, in terms of baselines, where I found the "+guidance" baselines particularly useful, and in terms of ablation studies performed.

**Weaknesses:**

The work presents some weaknesses, other than the limitations presented by the authors:
* **Problem**: the main problem addressed by this work (how to "stylize behaviours" given context information) is interesting but it's somehow narrow. Nonetheless, the method seems to perform better also in settings where context is not provided
* **Prompting**: as many works employing LLMs, the way the LLM is prompted seems to be crucial for correctly guiding the diffusion process. This means that changing the LLM, a new specific prompt structure would be required

**Questions:**

* what is the current inference time and how does it compare with the other approaches?
* line 461 multimodal, should be multi-modal to align with the rest of the text
* typo "Planing", in Appendix

**Limitations:**

Limitations are described in the paper, though the authors may consider adding some of the Weaknesses described above (e.g. prompting).

---

> ### Author Rebuttal · Authors · 2024-08-06
>
> ## [Weakness 1] Problem Definition
> We believe our work addresses a novel practical problem where the agent adapts to unseen language-specified contexts in a zero-shot manner, even when trained without a context-labeled dataset. Previous works on language-conditioned skill learning [1,2,3] primarily focus on learning through direct supervision from language-annotated datasets. However, obtaining datasets annotated with a broad range of contexts is impractical in real-world scenarios due to the open-ended nature of language. Furthermore, in the realm of RL, while diffusion-based policies have been explored primarily for imitating given datasets [4,5], language-guided test-time control for zero-shot adaptation has not been thoroughly investigated. To the best of our knowledge, our LDuS is the first to integrate LLMs with diffusion models for zero-shot adaptation to language-specified contexts in the domain of sequential decision-making.
>
> [1] Garg, Divyansh, et al. "LISA: Learning interpretable skill abstractions from language." NeurIPS 2022
>
> [2] Zhang, Edwin, et al. "Language control diffusion: Efficiently scaling through space, time, and tasks." ICLR 2024
>
> [3] Chen, Lili, et al. "Playfusion: Skill acquisition via diffusion from language-annotated play." CoRL 2023
>
> [4] Wang, Zhendong, et al. "Diffusion policies as an expressive policy class for offline reinforcement learning." ICLR 2024
>
> [5] Chen, Chang, et al. "Simple hierarchical planning with diffusion." ICLR 2024
>
> ----
>
> ## [Weakness 2] LLM Prompting
> In order to obtain better responses from LLMs, it is widely known that the prompt should be optimized for each LLMs. However, our primary focus is on addressing the challenge of zero-shot skill adaptation in language contexts, not on developing prompting methods for LLMs. Therefore, there remains room for improving our prompting approach.
>
> ----
>
> ## [Question 1] Inference Time
> In the table below, we present the average inference time (in milliseconds) required per timestep for LDuS and the baselines in MetaWorld. For comparison, we also report LDuS-Guidance, which is LDuS without the loss guidance. As shown, LDuS exhibits inference times similar to the baselines when only considering the diffusion sampling time. However, LDuS requires additional time for LLM inference. The LLM inference time can be further reduced by distilling essential knowledge, such as code generation and verification capabilities, into a smaller language model.
>
> |Model|Diffuser|LCD|LDuS-Guidance|Diffuser+Guidance|LCD+Guidance|LDuS|
> |---|---|---|---|---|---|---|
> Inference Time|55ms|56ms|56ms|102ms|100ms|108ms (Diffusion Sampling) + 55ms (LLM)|
>
> ----
>
> ## [Question 2 & 3] Typo
> We thank the reviewer for identifying the typos. We will correct these typos in our final version.

---

> > ### Comment · Reviewer_261B · 2024-08-12
> >
> > I am satisfied with the author's rebuttal.
> >
> > I would recommend including the inference time comparison in the paper, as this shows one of the current limitations of the approach.
> >
> > I will keep my score and recommend acceptance of the work.

---

> ### Author Response · Authors · 2024-08-13
>
> We are very encouraged by your satisfaction with our rebuttal. As addressed in response to question 1, we will include the inference time comparison in our paper. We deeply appreciate for your thoughtful feedback, as well as your decision to the recommend acceptance of our work.

---

### Official Review · Reviewer_UfZc · 2024-07-26

**Soundness:** 4
**Presentation:** 4
**Contribution:** 2
**Rating:** 7
**Confidence:** 5

**Summary:**

This paper presents an LLM-based policy adaptation framework for a language specified context. Here, context is a slight variation in how the task is performed. It has two stages, in skill-learning phase, a skill-based diffusion policy is learned with in-painting technique. Here, skill is a VAE encoding of a sequence of states-actions. In adaptation phase, the generation process of diffusion policy is guided with loss-guided diffusion (LGD) mechanism, where loss is generated by LLM conditioned on the context. Experimental results generally show that their skill-based policy along with LLM-generated-LGD is better at context following that other baselines.

**Strengths:**

This paper does a good job of combining LGD's usefulness with controlled generation and capabilities of LLMs to generate context following loss functions for robotic manipulation problem. The experiments/ablations are exhaustive and comparison with diffuser validates most key design choices like skill-conditioning, iterative-refining, in-painting etc). The paper is clear and coherent. Overall, it's a good proof of concept of how knowledge in LLMs can be distilled at test-time in zero-shot manner for controlled generation.

**Weaknesses:**

1. The scope of "context" is only limited to the agent's speed in the experiments. Meaning, the only type of adaptation/user-requirement studied in the paper is speed variation. Generating loss function for this is an easy task for an LLM given we have already seen objective generation on much harder tasks in works like [Eureka] and [Lang-to-Rew]. Perhaps experimenting with other types of context would have offered better insight into scalability of such approaches.
2. The methods has only been shown to work with low-dimensional state and not with images which raises question on it's real-world applicability. Although for given contexts, proposed framework might work with image-action data (as actions are sufficient to define trajectory for loss generation by LLM) but proof of that is missing. Perhaps an ablation where no low-dimensional state is used elsewhere except in skill-encoder and skill-prior, with only actions in trajectory, could have been performed to study the reliance of performance on state-info in both stages.

[Eureka]: https://arxiv.org/abs/2310.12931
[Lang-to-Rew]: https://arxiv.org/abs/2306.08647

**Questions:**

1. Line 197-198: How's error in the loss function identified if true loss function is not known? Line 638 mentions "if LDuS detects errors in the loss function", how is this detected? If this step is not robust then how will it be sure if mismatch is due to loss function or due to guidance weight?
2. Line 207: If speed variation is already captured in the training data then how is the evaluation ensured too be zero-shot? Details about the range of speed values during data collection and evaluation should be mentioned. t-SNE figure does show OOD generation but does not imply if all evaluation instances were OOD.
3. LDuS's no-iterative-refinement version outperforms diffuser suggesting skill-based diffusion is contributing a lot to the performance gain. In this regard, [Skill-diffuser] is a more relevant baseline to compare as it also conditions the action diffuser on latent skill computed using lang and image obs. It would have been a more fairer comparison in skill-based category as compared to LCD, as LCD is diffusion in latent skill space with RNN action head. Do you agree with this? If no, why? If yes, why wasn't this baseline compared?

[Skill-diffuser]: https://arxiv.org/abs/2312.11598

**Limitations:**

Image-based policy experiments are important to ensure real-world applicability of robotic methods and their abscence should have been addressed.

---

> ### Author Rebuttal · Authors · 2024-08-06
>
> ## [Weakness 1] Scope of Context
> We thank the reviewer for the thoughtful and constructive feedback. As the scope of context is limited in our experiments, we conduct additional experiments on two different types of user requirements including energy and spatial. In the energy context, the agent aims to minimize its energy consumption by reducing its acceleration or deceleration [1]. In the spatial context, the agent is tasked to not cross a specified spatial boundary. In the table below, we conduct tests on a single task in MetaWorld. As shown, LDuS outperforms the baselines in context reward (CR), demonstrating its scalability on diverse context types.
>
> |Method|Energy Context (CR)|Energy Context (SR)|Spatial Context (CR)|Spatial Context (SR)|
> |---|---|---|---|---|
> |LCD+Guidance|56.3|66.7\%|29.4|33.3\%|
> |Diffuser + Guidance|62.3|66.7\%|75.4|100.0\%|
> LDuS|88.8|100\%|87.0|100.0\%|
>
> Moreover, for complex tasks such as dexterous control, interactions with the environment [2] or API functions [3] are necessary. This is because LLMs are not inherently grounded on these complex environments, making them challenging to  generate loss functions directly. Therefore, we believe that incorporating such approaches (Eureka and Lang-to-Rew) will enable LDuS to accommodate more complex tasks.
>
> [1] Soori, Mohsen, et al. "Optimization of energy consumption in industrial robots, a review." Cognitive Robotics 2023
>
> [2] Ma, Yecheng Jason, et al. "Eureka: Human-level reward design via coding large language models." ICLR 2024
>
> [3] Yu, Wenhao, et al. "Language to rewards for robotic skill synthesis." CoRL 2023
>
> ----
>
> ## [Weakness 2 & Limitation] Image-based Experiment
> Our LDuS framework can be seamlessly extended to accommodate image inputs, and we conduct an additional experiment in a single task in MetaWorld. To handle image inputs, we first train a ResNet-based autoencoder to obtain the image encoder. Subsequently, we use the image embeddings produced by the encoder as the state for training LDuS,  without using any low-dimensional states.
>
> In the below table, LDuS+Img denotes the image-based version of LDuS. As shown, LDuS+Img exhibits only a slight performance drop compared to LDuS, demonstrating the applicability for image-based experiment setups. Additionally, there is certainly room for improvement of LDuS+Img, as we do not have enough time to optimize the hyperparameters or test with larger network sizes. We will include these experiments in the final version of our paper.
>
> |Method|Without Context (SR)|Language Context (CR)|Language Context (SR)
> |---|---|---|---|
> LDuS |100.0\%|92.7|100.0\%|
> LDuS+Img|83.3\%|87.5|83.3\%|
>
> ----
>
> ## [Question 1] Error Detection in Loss Function
> For errors due to the loss function, we employ self-critic mechanism of LLMs. This involves prompting the generated code again into the LLM to verify its alignment with the given context, and this verification occurs only at the initial. For error due to the guidance weight, the LLM evaluates whether the guided trajectory requires further modification, and if so, increases the frequency of loss guidance application. This verification is executed for every skill horizon. As mentioned in the response to the first question, for complex tasks, grounding the LLM to the environment is necessary to ensure robust error detection in the loss function.
>
> ----
>
> ## [Question 2] Experiment Settings for Zero-shot Adaptation
> We set the target contexts for evaluation to those not present in the training dataset. For example, in the button press task, the training dataset comprises speeds of $[0.22, 0.27, 0.31, 0.34, 0.37]$,  and we test speeds of $[0.32, 0.33, 0.38, 0.40]$ for zero-shot adaptation. As we set different speed configurations for each task, we will include the details of our experiment settings in our final version.
>
> ----
>
> ## [Question 3] Skill-diffuser Baseline
> Given the similarities in the overall skill learning structure between Skill-diffuser and LCD, we deem it sufficient to compare only one as the baseline. Instead, we choose to compare with Diffuser and LangDT, which employ different structural approaches, such as decision diffusion-based planner and decision transformer, to highlight the differences between such design choices. Moreover, as the Skill-diffuser also focuses on learning skills by direct supervision from the datasets, we conjecture that it may not be capable of zero-shot adaptation to contexts. We will include the Skill-diffuser as a baseline for our final version.

---

> > ### Comment · Reviewer_UfZc · 2024-08-10
> >
> > I thank authors for running additional experiments, especially the spatial context ones, it definitely builds more confidence into the method. I believe the proposed work is a promising direction towards incorporating various user-contexts and safety constraints. All my concerns have been addressed and I have updated my score.

---

> > > ### Author Response · Authors · 2024-08-11
> > >
> > > We are very satisfied that your concerns have been addressed and appreciate your decision to increase the score. We are also greatly encouraged by your recognition of our work as promising.
> > >
> > > For the final version of our paper, we will include experiments with the 'spatial' and 'energy' contexts, as well as the image-based version of LDuS. Additionally, we will include details of our zero-shot evaluation settings.
> > >
> > > Once again, we truly appreciate your thoughtful and insightful feedback.

---

### Author Rebuttal · Authors · 2024-08-06

We deeply appreciate the valuable and constructive feedback from the reviewers. We addressed all identified weaknesses and questions to resolve the concerns raised. For those that require additional experiments, we have conducted further studies to provide a more comprehensive understanding. Lastly, we provide mathematical justification for Equation (5) in the attached PDF.

---

### Decision · Program_Chairs · 2024-09-25

**Decision:**

Accept (poster)

**Comment:**

This paper presents an LLM-based policy adaptation framework for language-specified context in robotic tasks.
It combines a skill-based diffusion policy learned with in-painting, All reviewers agree that this is an innovative integration of LLM reasoning and loss-guided diffusion, and the paper demonstrates robustness across context types, with a clear presentation. The approach shows good performance  in skill-based learning task with context-specific instructions, particularly for speed variations in robotic manipulation tasks. Although, limitations exist in prompt dependency.

However the rebuttal did a good job of providing clarifications with more than one reviewers updating the scores leaning more positively.  All reviewers lean accept after rebuttal phase, and the Metra-reviewer concurs on the overall agreement. .
The authors are advised to review the final comments, and update the manuscript accordingly.. In addition to including clarifications in the main paper, the authors are also advised to describe the possible limitations of this work in more detail.